# Relationship between family functioning, differentiation of self and anxiety in Spanish young adults

**Blanca Dolz-del-Castellar[1], Jesús Oliver[1,2]***

**1** Department of Psychology, Comillas Pontifical University, Madrid, Spain, **2** Department of Personality, Assessment and Psychological Treatment, University of Málaga, Málaga, Spain

* jesus.oliver@uma.es

## Abstract

### Objectives

In the present study, the relationship between family functioning, differentiation of self and trait anxiety was examined. In addition, differences in trait anxiety and differentiation of self according to sex were studied. It was also checked whether family functioning predicted the achieved degree of differentiation of self, and whether family functioning and differentiation of self predicted the level of trait anxiety. Finally, it was analyzed whether the level of differentiation of self mediated the relationship between family functioning and trait anxiety.

### Methods

The study involved 185 Spanish participants, aged between 18 and 56 years and the scales used were the Family Adaptability and Cohesion Evaluation Scale-20Esp (FACES-20Esp), the Differentiation of Self Scale (EDS) and the State-Trait Anxiety Inventory (STAI).

### Results

The results indicated that family functioning was related to differentiation of self and trait anxiety, and that differentiation of self was related to trait anxiety. In addition, according to sex, differences were found in the degree of differentiation of self and in the I Position, with a higher score for men, as well as in Emotional Reactivity and trait anxiety, with higher scores for women. It was also observed that family functioning predicted the level of differentiation of self, and that differentiation of self and family functioning predicted trait anxiety. Finally, it was found that the relationship between family functioning and trait anxiety was partially mediated by differentiation of self.

### Conclusions

There are relationships between family functioning, differentiation of self and trait anxiety, and there are differences in differentiation of self and trait anxiety based on sex. The relevance of the results and their implications for clinical practice are discussed.

**Data Availability Statement:** All relevant data are within the paper and its Supporting Information files.

**Funding:** The study was supported by the Universidad de Málaga (ES), received by JO. The

funders had no role in study design, data collection and analysis, decision to publish, or preparation of the manuscript.

## Family functioning

The Circumplex Model [1] describes the family functioning through three dimensions that have been considered of great relevance by different family theoretical models and family therapy approaches: Cohesion, Adaptability and Communication [2].

Cohesion refers to the emotional bond that the members of a family system have with each other [1,3]. Adaptability is defined as the ability of a system to change its power structure, the roles and the relationship rules in response to a specific stressor or to changes that occur due to the system development [3]. Communication is a facilitating dimension, that promotes that systems move between the other two dimensions [3].

The dimensions of Cohesion and Adaptability are curvilinear, that is, the ends of both dimensions are dysfunctional, while the central positions are considered related to an adequate functioning [1,4]. However, the different versions of the Family Adaptability and Cohesion Evaluation Scale, except version IV, evaluate them in a linear way. That is, the greater family cohesion and adaptability, the better family functioning.

Different studies have shown that the adequate family dynamics in the family of origin decreases the likelihood of future behavior problems in children [5,6]. Likewise, it has been found that family patterns, especially those related to the regulation of distance between members of the system, have a significant impact on trait anxiety in children [7,8]. In this way, it has been found that higher levels of cohesion and lower levels of adaptability (rigidity) in the family are associated with higher levels of social anxiety in children [9] and lower levels of psychological flexibility and self-compassion [10].

In a research study with English young people, Manzi et al. [11] found that the family fusion (enmeshment), which means very high levels of Cohesion where the boundaries between family members are diffuse, and rigid with the outside, was associated with more anxious and depressive symptoms. Sawyer et al. [12] conducted a study with African-American adults, in which they found that greater family dysfunction in childhood was associated with more symptoms of anxiety and depression in adulthood.

On the other hand, in the study previously mentioned, Manzi, et al. [11] found that an adequate family cohesion was positively associated with life satisfaction and negatively associated with depressive and anxious symptoms. In the same way, Guassi et al. [13] found that those young people who indicated higher levels of family cohesion (measured in a linear manner) during the transition to university, had lower levels of depressive symptoms.

Likewise, Uruk et al. [14] found that family functioning (Adaptability and Cohesion) was positively associated with psychological wellbeing. Davenport et al. [15] also observed that close and flexible family relationships were related to a lower perception of stress in African-American adults. In addition, Alavi et al. [16] pointed out that higher scores in Adaptability and Cohesion are positively related to higher levels of emotional intelligence in young people.

In line with Bowen's theory, Johnson et al. [17] found that family functioning affected the levels of differentiation of young adults from families with divorced parents. That is, not only the divorce was negatively related to the differentiation of self, but family cohesion seemed to reduce Emotional Reactivity, while family conflict seems to affect the level of differentiation, increasing Emotional Reactivity and Emotional Cutoff and decreasing I Position. Likewise, Chung and Gale [18] found that family functioning was positively related to differentiation of self, in such a way that the greater the degree of perception of family functioning as healthy, the greater the degree of differentiation.

In summary, both theory and research suggest that experiences lived in the family have a central role in the adjustment and emotional development of young adults [19,20].

## Differentiation of self

Differentiation of self is defined as the ability to maintain the balance between emotional and intellectual functioning and between the degree of intimacy and autonomy in the relationships [21,22]. At the intrapsychic level, differentiation of self consists of the ability to distinguish between intellectual processes and emotional processes, that is, it is the ability to distinguish thoughts from feelings and to be able to choose to act in accordance with ones or the others [22,23]. At the interpersonal level, differentiation of self refers to the ability of an individual to establish intimate relationships with others while being able to maintain his autonomy [22].

People who have lower levels of differentiation of self in the intrapsychic component find it more difficult to remain calm before the emotionality of [7] and are less able to modulate their emotional activation [24], what hinders their capacity to assess reality in a more balanced way and to reflect on it, to deal more adequately with life stressors and to tolerate ambiguity and uncertainty [25]. Those people who present lower levels of differentiation in the interpersonal component tend to fusion with others, to be dominant with others or to distance themselves emotionally, through the emotional cutoff, as a response to stress or a way to handle it [23,26].

Theoretically, differentiation of self is composed of five dimensions: Emotional Reactivity, I Position, Fusion with Others, Emotional Cutoff [22] and Dominance over the Others [26].

Emotional Reactivity is the inability to remain calm in the presence of significant people who are emotionally activated and the tendency to respond through an automatic emotional response to environmental stimuli [22]. I Position has been defined as the ability to maintain a clear sense of self, as well as one's own convictions and beliefs against significant others when one is pressed to do otherwise [22]. Fusion with Others is defined as the tendency to seek union with significant others and the approval and acceptance of others above all [22]. Emotional Cutoff is the tendency to isolate oneself, to maintain independence with others and to cut off relationships with significant others, in a rigid manner, as a way of managing tension and intimate relationships [22]. Finally, Dominance over the Others, refers to a low capacity to tolerate differences with others and a propensity to pressure others to conform to the own ideas or interests, entering power struggles or becoming inflexible or dogmatic [26].

Bowen postulated that the level of differentiation of self in each individual develops during childhood, is consolidated throughout adolescence and remains relatively stable for the rest of the individual's lifetime, although some later life experiences, or a structured effort to increase it can generate changes on the differentiation level [21].

The degree of differentiation of self that individuals achieve is strongly related to their parent's level of differentiation. Through a process that the author called "family projection", parents transmit to their children their anxiety, their ability to distinguish the emotional and intellectual system, their ability to interact without losing their autonomy and, ultimately, their level of differentiation. As parents tend to project more in some children than others, differences between children in their level of differentiation usually occur [21].

Bowen hypothesized that, although the level of differentiation of self a person reaches has no direct relationship with the presence or absence of symptoms, the most differentiated people tend to experience lower levels of chronic situational anxiety, fewer physical and psychological symptoms, and less social problems than individuals who are less differentiated, especially during particularly stressful periods [21].

Several investigations have pointed out that lower levels of differentiation are associated with higher levels of chronic anxiety, higher levels of psychological suffering [22,27], lower levels of marital satisfaction [22,28], higher levels of social anxiety [29,30] and more physiological [29,31] and psychological symptoms [22,27,31]. Higher levels of differentiation have been

found related to greater psychological and relational well-being [32], lower symptomatology and social anxiety [29,33] and better coping [24,33].

Chung and Gale [34], meanwhile, found that differentiation of self in both Korean and American young adults was related to psychological well-being. Specifically, they found that lower levels of Emotional Cutoff and greater ability to take an I Position were related to higher levels of self-esteem and lower levels of depression. In a similar way, Jenkins et al. [35] found that people with greater Emotional Reactivity trust less in their own abilities and have a less stable identity, while being able to adopt an I Position is essential for a good psychosocial development in young adults.

Other authors [36], in a research carried out with Korean population, found that people with higher levels of differentiation, referred to have healthier family functioning, greater family satisfaction and more positive family communication. More specifically, they found that I Position had a significant effect on the balanced levels of Cohesion and Adaptability and that those with lower levels of Emotional Cutoff had more balanced levels of Cohesion [36]. These same authors [37], in a study carried out with South Koreans living in South Korea, South Koreans living in the United States and white Americans living in the United States, also found that higher levels of differentiation of self were significantly related to a healthier family functioning, better family communication and greater family satisfaction in the three groups.

## Anxiety

Anxiety can be defined as the response of the organism to a threat, real or imagined [21]. The different alterations that occur in the individual due to anxiety can be of three types: physiological, cognitive and motor [38]. The responses of anxiety have an important adaptive function for the organism; however, if these reactions occur too intensely or at times when they are not necessary, this capacity for adaption may be impaired [21,39] and generate great discomfort. According to Spielberger et al., [40], there are two main types of anxiety: trait anxiety and state anxiety. The first refers to the relatively stable tendency of an individual to consider or perceive a wide variety of stimuli as dangerous or threatening, while the second refers to a state of transitory or punctual emotional activation that varies its intensity or changes over time or depending on the stimuli.

In the scientific literature, the context, especially the family, has been pointed as an important indicator of psychological adjustment and emotional well-being in young students [41]. Specifically, anxiety is one of the variables that has been most associated in research with family dysfunction [42].

Several studies have found anxiety is negatively related to academic performance [43], self-esteem [39], substance abuse [44], and positively related to physical, social, functional and emotional well-being [45].

In line with Bowen's theory, some investigations have found that differentiation of self is also related to trait anxiety [22,29,30,46,47].

As we have seen, a large number of studies points out the relationship between family functioning and various psychological variables, and the importance they have in the development of the individual. However, little attention has been paid to the relationship between family functioning and differentiation of self, both in international and, above all, in Spain, despite the importance given to these constructs and the repercussions they have in the clinical field. Furthermore, the study of family functioning, differentiation of self and trait anxiety in Spain may be very interesting, because families are more collectivistic and prioritize the mutual support and interdependence [46,48]. For these reasons, we consider it relevant to carry out an investigation in Spain, in which the empirical relationship between these theoretically related variables is studied.

The objectives of this study are: a) to explore the relationship between family functioning, differentiation of self, and trait anxiety in a Spanish sample; b) to observe if there are differences in differentiation of self and anxiety based on sex; c) to check whether family functioning and differentiation of self predict anxiety; d) to test whether family functioning and differentiation of self predict anxiety; and e) to analyze if the relation between family functioning and anxiety is mediated by differentiation of self.

## Method

### Research ethics statement

The Ethics Committee of the Comillas Pontifical University of Madrid approved the study. The data collection was done through an online questionnaire, made in Google Forms and sent through different platforms. This included a presentation with the name of the researchers, the purpose of the study, the commitment of confidentiality, the instructions and the gratitude for the participation. Before filling in the questionnaire, the participants had to give their written consent.

### Sample

The sample was collected through a non-probabilistic sampling of convenience. The subjects were Spanish students of the Universidad Pontificia Comillas, who were asked to participate in the study. In addition, a snowball procedure was used, requesting participants to propose other acquaintances to participate in the study as well.

In the present study 208 subjects participated, of which 23 were eliminated since they did not meet any of the inclusion criteria: being of legal age (18 or older), not having a family of their own and living with their family of origin at the time of the study.

Of the 185 subjects that made up the final sample, 123 were women (66.5%) and 62 men (33.5%). Their ages were between 18 and 56 years old ($M$ = 22.08, $SD$ = 3.78). Of the total sample, 20% have postgraduate studies, 59.5% university studies, 6.5% studies in Vocational Training and 14% High School Senior year. 83.8% of the total sample lives in a two-parent family, 11.9% in a single-parent family and 4.3% in a reconstituted family. Of those who live in a single-parent or a reconstituted family, 46.7% do it due to the divorce of the parents and 30% to the death of one of the parents.

### Instruments

**a) Sociodemographic questionnaire.** An ad hoc questionnaire was designed to assess various sociodemographic variables such as age, sex, marital status, educational level, family typology, divorce or death of one of the parents, current coexistence with the family of origin, existence of own children, and reception or not of psychotherapy at some time.

**b) Differentiation of Self Scale.** The Differentiation of Self Scale (DSS; in Spanish, Escala de Diferenciación del Self, EDS), developed by Oliver and Berástegui [26], is an instrument that assesses the Differentiation of Self, both the intrapsychic and interpersonal dimensions. The scale consists of 74 items that are answered by a Likert-type scale with six response options, ranging from 1 (strongly disagree) to 6 (strongly agree) and grouped into five subscales: Emotional Reactivity (ER; 12 items), Emotional Cutoff (EC; 21items), Fusion with Others (FO; 14 items), Dominance over the Others (DO; 14items) and I Position (IP; 13items). The total score of the scale and subscales range between 1 and 6. The relationships between the subscales FO, DO, ER and EC and the total scale of differentiation are inverse, while the relationship between the subscale IP and the total scale is direct.

This instrument presents high internal consistency indices (Cronbach's alpha), both for the total scale (total DSS = .93) and for the different subscales (ER = .89; IP = .86; FO = .90; DO = .89; EC = .90). With respect to the construct validity, the exploratory factor analysis showed a structure of five factors, which explained 45% of the variance, as established by the authors. Regarding other validity evidences, significant relationships were found between differentiation of self and its dimensions with different theoretically related variables such as family satisfaction, emotional maturity, degree of general well-being and the usual level of anxiety. In the present study, good internal consistency indices were also obtained, both for the full scale, which obtained a Cronbach's alpha of .86, and for the subscales (ER = .88; IP = .82; FO = .86; DO = .86; EC = .88).

**c) Family Adaptability and Cohesion Evaluation Scale-20Esp.** This instrument is a Spanish adaptation of the Family Adaptability and Cohesion Evaluation Scale II (FACES II) [49], carried out by Martínez-Pampliega et al. [4], that assesses family functioning. The FACES-20Esp is composed of 20 items, with a Likert-type scale with scores ranging from 1 (never or almost never) to 5 (almost always). The items are grouped into two subscales, of ten items each, which measure two dimensions of family functioning: adaptability and cohesion [1,3]. Although the Circumplex Model considers these two dimensions as curvilinear, this questionnaire measures them linearly.

This instrument has obtained high internal consistency indices (Cronbach's alpha): .89 in Cohesion and .87 in Adaptability. The construct validity was supported by a confirmatory factor analysis that indicated the bidimensional structure established by the authors. Regarding other validity evidences, it was found that this scale was congruently associated with other instruments that measure related constructs, such as the Family Environment Scale. In the present study, high internal consistency indices were obtained (Cronbach's alpha): Cohesion = .92, and Adaptability = .91.

**d) State-Trait Anxiety Inventory.** The State-Trait Anxiety Inventory (STAI), developed by Spielberger et al. [50], has been adapted to Spanish by Seisdedos [51]. This questionnaire is composed of two scales that measure two different constructs: Trait Anxiety (T/A) and State Anxiety (S/A). Each of the scales is composed of 20 items that are answered with a Likert-type scale ranging from 0 (nothing) to 3 (a lot) for the S/A scale, and from 0 (almost never) to 3 (almost always) for the T/A scale. The total score of each scale, which can oscillate between values from 0 to 60, is obtained by adding the values of the different items that make it up, in such a way that the highest scores correspond to higher levels of anxiety [52].

The psychometric properties of this instrument are adequate. In the validation studies showed an internal consistency (Cronbach's alpha) between .83 and .92. In addition, it presented evidence of concurrent validity by relating the STAI measure to that of other questionnaires that also measure anxiety [51]: the Affect Adjectives Check List (AACL) [53], the Institute of Personality and Ability Testing-Anxiety Scale (IPAT-AS) [54], and the Taylor's Manifest Anxiety Scale [55]. In the present study, only the trait anxiety scale was used, with which a Cronbach's alpha of .91 was obtained.

## Procedure

The data collection was done through an online questionnaire, made in Google Forms and sent through different platforms. This included a presentation with the name of the researchers, the purpose of the study, the commitment of confidentiality, the instructions and the gratitude for the participation. Data from people who met all the inclusion criteria were entered into the SPSS-21 and subjected to pertinent statistical tests.

## Data analysis

A series of statistical analysis were carried out to meet the objectives of the study. To determine the appropriate statistical test in each case, the assumptions for each of the tests were checked. The Pearson´s r tests were used to analyze the relationships between the variables of the study. The Student´s t tests for independent samples were carried out to analyze differences of means in differentiation of self and anxiety based on sex. After checking the pertinent assumptions, a simple linear regression was performed to check if family functioning predicts the level of differentiation, and a multiple linear regression was conducted to explore if family functioning and differentiation of self predict anxiety. Also, a partial correlation analysis was carried to test the effect of the differentiation of self on the relationship between the family functioning and anxiety.

Finally, to check if the differentiation of self mediates the effect of family functioning on trait anxiety, a mediation analysis was performed using a bootstrapping procedure [56] with the macro PROCESS version 3 for SPSS [57]. The parameter estimates were based on 5000 bootstrap samples. The mediational model 4 was used, in which family functioning was considered as an independent variable, differentiation of self as a mediating variable and trait anxiety as a dependent variable. Confidence interval for the indirect effect was analyzed, and it was considered that if the interval did not include zero it would indicate a statistically significant indirect effect with $p < .05$ [58].

## Results

First, several Pearson's correlation tests were applied in order to contrast the relationship between family functioning and differentiation of self, family functioning and trait anxiety, and differentiation of self and trait anxiety was examined.

As can be seen in Table 1, the results showed moderate significant relationships between family functioning and DSS ($r = .43$, $p < .001$, $r^2 = .19$) and adaptability and DSS ($r = .43$, $p < .001$, $r^2 = .19$), and cohesion and DSS ($r = .39$, $p < .001$, $r^2 = .15$). In addition, significant relationships were found between the dimensions of the family functioning and the dimensions of the differentiation of self. Specifically, adaptability was moderately and negatively related to EC ($r = -.50$, $p < .001$, $r^2 = .25$) and slightly related to FO ($r = -.19$, $p = .009$, $r^2 = .04$), IP ($r = .22$, $p = .002$, $r^2 = .05$) and ER ($r = -.27$, $p < .001$, $r^2 = .07$). It was also observed that cohesion was moderately associated with EC ($r = -.52$, $p < .001$, $r^2 = .27$) and slightly related to IP ($r = .21$, $p = .005$, $r^2 = .04$) and ER ($r = -.20$, $p = .006$, $r^2 = .04$).

**Table 1. Matrix correlations between family functioning, differentiation of self and trait anxiety.**

|  | Family functioning | Adaptability | Cohesion | T/A |
|---|---|---|---|---|
| DSS | .43** | .43** | .39** | -.69** |
| ER | -.25** | -.27** | -.20** | .67** |
| IP | .23** | .22** | .20** | -.36** |
| FO | -.17* | -.19** | -.12 | .33** |
| DO | -.14 | -.12 | -.14 | .12 |
| EC | -.54** | -.50** | -.52** | .55** |
| T/A | -.44** | -.41** | -.42** | — |

*Note*. DSS = Differentiation of Self Scale; ER = Emotional Reactivity; IP = I Position; FO = Fusion with Others; DO = Dominance over the Others; EC = Emotional Cutoff; T/A = Trait Anxiety.

*$p < .05$

**$p < .01$

It was also observed moderate negative relationships between family functioning and T/A ($r = -.44$, $p < .001$, $r^2 = .19$), as well as between adaptability and T/A ($r = -.41$, $p < .001$, $r^2 = .16$) and between cohesion and T/A ($r = -.42$, $p < .001$, $r^2 = .17$).

Likewise, it was found that DSS and T/A are negatively and rather highly related ($r = -.69$, $p < .001$, $r^2 = .48$). Finally, significant and positive relationships were found between RE and T/A ($r = .67$, $p < .001$, $r^2 = .45$), with a high magnitude; CE and T/A ($r = .55$, $p < .001$, $r^2 = .30$), with a moderated magnitude; FO and T/A ($r = .33$, $p < .001$, $r^2 = .11$); and IP and T/A ($r = -.36$, $p < .001$, $r^2 = .13$), with a negative relationship.

Next, to check if there are differences in the differentiation of self and its dimensions and in trait anxiety according to sex, a Student's t test for independent samples was applied.

As shown in Table 2, moderate differences were found in DSS according to sex ($t(183) = 2.24$, $p = .026$, *Cohen´s d* = .35), with a higher mean in the group of men ($M = 4.33$, $SD = .41$) compared to the group of women ($M = 4.17$, $SD = .49$). Furthermore, large differences were observed between men and women in ER ($t(183) = -4.88$, $p = < .001$, *Cohen´s d* = .76), with higher scores in women ($M = 3.61$, $SD = .95$) than in men ($M = 2.92$, $SD = .84$); and moderate differences were found in IP ($t(183) = 2.37$, $p = .019$, *Cohen´s d* = .37), with higher scores in men ($M = 4.77$, $SD = .59$) than in women ($M = 4.54$, $SD = .63$). Finally, statistically significant differences were found in T/A according to sex ($t(183) = -2.84$, $p = .005$, *Cohen´s d* = .45), with a higher mean in women ($M = 25.67$, $SD = 11.39$) than in men ($M = 20.79$, $SD = 10.28$), and a moderate effect size.

To check if the family functioning predicts the level of differentiation of self achieved and if family functioning and differentiation of self predict anxiety, two linear regression analysis were performed. Before, it was observed that the relationship between Cohesion and Adaptability was high ($r = .79$, *Cronbach's alphas of both subscales* = .95). To solve the problems of multicollinearity, the total family functioning score was included as a predictor variable, instead of its two dimensions. A simple linear regression revealed that family functioning ($\beta = .43$, $p < .001$) explained 18.8% of the DSS scores ($R^2 = .188$, $F(1,183) = 42.29$, $p < .001$). Furthermore, a multiple linear regression revealed that family functioning ($\beta = -.17$, $p < .001$) and differentiation of self ($\beta = -.62$, $p < .001$) explained 50.3% of trait anxiety ($R^2 = .503$, $F(2,182) = 91.95$, $p < .001$).

Finally, in order to contrast if the relationship between family functioning and trait anxiety is mediated by the differentiation of self, a partial correlation analysis was first applied to verify

**Table 2. Difference of means between women and men in differentiation of self and trait anxiety.**

|  | Women ($n = 123$) | | Men ($n = 62$) | | | | |
|---|---|---|---|---|---|---|---|
|  | *Mean* | *S.D.* | *Mean* | *S.D.* | *t* | *p* | *Cohen´s d* |
| DSS | 4.17 | 0.49 | 4.33 | 0.41 | 2.24* | .026 | .35 |
| ER | 3.61 | 0.95 | 2.92 | 0.84 | -4.88** | < .001 | .76 |
| IP | 4.54 | 0.63 | 4.77 | 0.59 | 2.37* | .019 | .37 |
| FO | 2.70 | 0.79 | 2.52 | 0.62 | -1.66 | .099 | .24 |
| DO | 2.64 | 0.75 | 2.85 | 0.62 | 1.83 | .069 | .28 |
| CE | 2.74 | 0.81 | 2.83 | 0.70 | 0.70 | .482 | .11 |
| T/A | 25.67 | 11.39 | 20.79 | 10.28 | -2.84** | .005 | .45 |

*Note*. DSS = Differentiation of Self Scale; ER = Emotional Reactivity; IP = I Position; FO = Fusion with Others; DO = Dominance over the Others; EC = Emotional Cutoff; T/A = Trait Anxiety.

*$p < .05$

**$p < .01$

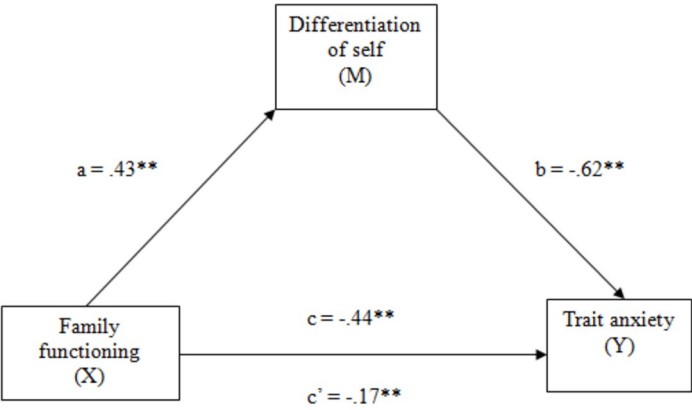

**Fig 1.**

the degree of linear relationship between family functioning and anxiety after eliminating the effect of differentiation of self. The partial correlation shows that there is a negative and moderate relationship between family functioning and T/A ($r = -.44$, $p < .001$, $r^2 = .19$), as well as the relationship between family functioning and T/A is significantly altered after controlling the effect of DSS ($r = -.21$, $p = .004$, $r^2 = .04$). Therefore, it was decided to do a mediation analysis. Finally, to check if the differentiation of self mediates the effect of family functioning on trait anxiety, a mediation analysis was performed. The results of the analysis (see Fig 1) indicate that the indirect effect is significant, since the confidence interval (*CI* 95%) does not include zero ($B = -.27$, *SE* $= .05$, 95% *CI* [-.36, -.18]). However, the data show that this is a partial mediation, since the effect of family functioning on trait anxiety remains significant.

## Discussion

The objective of this study was to examine the relationship between family functioning, differentiation of self and trait anxiety in a Spanish sample. In addition, it was studied if there were differences in differentiation of self and trait anxiety depending on sex. Furthermore, it was checked if family functioning predicted differentiation of self, and if family functioning and differentiation of self predicted trait anxiety. Finally, it was checked if differentiation of self mediated the relationship between family functioning and trait anxiety.

First, the results showed that there is a positive relationship between family functioning and differentiation of self, in such a way that the better the family functioning and the greater degree of Adaptability and Cohesion a family presents, the higher the level of differentiation of self presented by its members. These results are consistent with the Bowen's theory, which states that functional family dynamics favor the level of differentiation of its members and that more differentiated people have more functional relationships with others. In addition, they are in line with what was found by authors such as Chung and Gale [18] and Kim et al. [36,37], which show that a healthy family functioning is positively associated with the degree of differentiation of self.

Likewise, it was found that Emotional Reactivity, Fusion with Others and Emotional Cutoff were negatively related to family functioning while I Position did it in a positive way. These

results were expected, since authors like Johnson et al. [17] reveal that family cohesion seems to reduce Emotional Reactivity, while family conflict seems to increase the levels of Emotional Reactivity and Emotional Cutoff and decrease the level of I Position. However, no relationship was found between Dominance over the Others and family functioning, as would have been expected, especially between Adaptability and Dominance over the Others. It would have been expected because the first variable is understood as the ability of a system to change and adjust to new demands of the environment and it is related to assertiveness and negotiation capacity [1], and the second variable refers to the difficulty to tolerate differences with others and the tendency to impose points of view or ideas [26]. However, it is possible that the absence of relationship is explained because some of the participants who have a dominant profile have had flexible parents, with capacity to negotiate the rules with them, while others have had authoritarian parents, and they have ended up being as dominants as their parents.

A negative relationship was also found between family functioning and trait anxiety. These results also agree with Bowen's postulates that point out that family dynamics are related to the chronic anxiety of the family and each of its members. In addition, these results coincide with those indicated by authors such as Sawyer et al. [12] who found that a poor family functioning in childhood was significantly related to higher levels of anxiety and depression in adulthood. In the same way, Chapman and Woodruff-Borden [42] and Manzi et al. [11] found that family functioning was related to anxious symptoms in adolescents and young adults. Likewise, Uruk et al. [14] found that those people who came from families with higher levels of Cohesion, Adaptability and Communication, presented lower levels of stress. These data could lead to think that through healthy relationships in the family, individuals are able to learn ways to manage stress. However, other studies have not found a direct relationship between dysfunctional (chaotic-fused) family functioning and anxiety [10]. These differences in the results could be explained by the different scales used to assess family functioning and anxiety, making it difficult to compare the results of the different studies, as well as differences in the conception of what is an adequate family functioning according to the country in which it is evaluated. Therefore, and given that the studies are inconclusive, it is important to continue studying this relationship, although most studies seem to indicate that there is a relationship between family functioning, anxiety and symptomatology [12,59,60].

It was also found that the higher the levels of differentiation of self, the lower the levels of trait anxiety presented by a person. These results are consistent with the Bowen's theory that considers differentiation of self as an aspect of great relevance in the development of an adequate psychological well-being and considers that low levels of differentiation of self are associated with chronic anxiety [21]. These results are in the same line as those found by other authors. Skowron and Friedlander [22], Duch-Ceballos et al. [46], Isik and Bulduk [47] and Peleg-Popko [29,30] noted that higher levels of differentiation of self were related to lower levels of trait and state anxiety and social anxiety. Skowron et al. [32] also found that a greater differentiation of self predicted fewer psychological symptoms and social problems over time. In the same line, Rodríguez-González et al. [31] found that differentiation of self predicted psychological health.

In addition, the results indicate that there is a statistically significant relationship between several dimensions of the differentiation of self and trait anxiety. These results are similar to those found by Peleg and Zoabi [33], in which they found that there was a negative relationship between I Position and social anxiety and a positive relationship between the dimensions of Emotional Reactivity, Fusion with Others and Emotional Cutoff and scores on social anxiety. Likewise, Duch-Ceballos et al. [46] found that there was a relationship between the dimensions of the differentiation of self and anxiety.

On the other hand, statistically significant differences were found in the degree of differentiation of self according to sex. Men had higher levels of differentiation of self than women, similar to results obtained in other studies [17,61]. Otherwise, there are studies that have not found differences between men and women in the differentiation of self [31,46], so the results do not seem conclusive. However, some of these results must be interpreted with caution since they evaluate the differentiation of self using only two dimensions, Emotional Reactivity and Emotional Cutoff.

In addition, statistically significant differences were found between men and women in Emotional Reactivity, being greater in women. These results coincide with those obtained in previous research [22,23,46,61]. On the other hand, significant differences were found depending on sex in I Position, with higher levels in men. These results coincide with those found by Skowron and Schmitt [23], although most studies have found no association between both variables [46,62]. However, in the present study no differences were found according to sex in Emotional Cutoff. Despite Skowron and Friedlander [22] and Johnson et al. [17] found that men present higher levels of Emotional Cutoff than women, this result supports those obtained in other investigations [46,61]. Although the results on the degree of differentiation and its dimensions according to sex are not conclusive and should be explored in the future, the differences obtained in this study could be related to the gender roles with which men and women have been socialized. These roles tend to promote in women those aspects related to the emotional world and the expression of feelings of vulnerability, while in men, tend to favor the affirmation of themselves and the recognition of their own strengths and hinder the expression of the vulnerabilities, aspects that seem to be more related to I Position and Emotional Cutoff. In addition, the fact that no differences were found between men and women in Emotional Cutoff, could have to do with the inequality in the sample size of men and women and with the possibility that, having selected a convenience sample, it has been accessed a group of men who are more able to connect with their emotions because they have studied Psychology.

Furthermore, significant differences according to sex at the level of trait anxiety were found, with higher levels of anxiety in women than in men. These results coincide with those found in previous studies [51,52,63]. Again, these differences could be due to the gender roles with which women are socialized, which tend to assign them a greater number of functions related to care and maintenance of interpersonal relationships than to men. This could cause overload on women, coming sometimes to neglect their own needs, and feelings of little worth or failure to not comply with these mandates, which increase their anxiety levels.

The present study also found that family functioning predicted by 19% the level of differentiation of self. That is, those subjects who perceive their families with adequate capacity to change and emotionally linked are better able to differentiate themselves, regulate themselves emotionally, maintain intimacy with others, connect with their own emotions, defend their own opinions and tolerate differences with others. Despite these results, it has been more common in the literature to investigate how the degree of differentiation of self influences family functioning, finding equally significant results like those of Kim et al. [36] that indicate that the levels of I Position, Emotional Cutoff and Fusion with Others predict 32% of the variance in Cohesion and the levels of I Position and Emotional Cutoff explain 27% of the variance in Adaptability. The fact that both variables can be used as predictors is due to the fact that, probably, both variables influence each other and, therefore, the choice to perform analysis in one direction or another of the relationship depends on where the researcher places the focus of interest. However, from an educational and evolutionary point of view the parents or caregivers are who should guarantee an adequate family functioning, in which Cohesion and Adaptability are not neglected, which favors the development of their children. The parents are who,

through interaction with their children, should teach them to regulate themselves emotionally and behaviorally. Therefore, the results of this study are relevant, since they indicate that one fifth of the level of differentiation of the sample is predicted by the relationships that parents establish in the family system.

It was also found that differentiation of self and family functioning predicted 50.3% of the level of trait anxiety. That is, those subjects who are better able to differentiate themselves and who perceive that their families are more cohesive and adaptable, have a lower trait anxiety. These results agree with Bowen's postulates that point out that family functioning and the level of differentiation of self of a person are related to its level of chronic anxiety. Uruk et al. [14], after controlling gender and ethnicity, also found that family functioning predicted 7% of post-traumatic stress in young adults. Likewise, Pollok et al. [64] observed, after controlling the sociodemographic variables, that family functioning predicted 12% of stress in a sample of African American adults. On the other hand, Duch et al. [46] found that differentiation of self accounts for 61% of the variance of trait anxiety.

Finally, it was found that differentiation of self partially mediated the relationship between family functioning and trait anxiety. The interpretation of these results suggests that the association between family functioning and the levels of trait anxiety that a person presents are partially explained by the ability of each individual to differentiate him or herself. These results are consistent with the postulates of Bowen's theory according to which the degree of differentiation of self modulates the level of anxiety experienced by the members of a family system. Likewise, these results indicate the relevance of differentiation of self in the appropriate psychological adjustment of individuals. It is possible that in this work no total mediation has been found because there are other intrapsychic factors, in addition to differentiation of self, such as social network, relationship or identity, which could mediate between family functioning and anxiety.

These results add empirical support to other results carried out with a Spanish sample that indicate the importance of the differentiation of self in relation to psychological well-being [31] and provide data in line with what has been studied in other countries [24,34].

The present study has some limitations. First, it is necessary to indicate the type of sampling used: convenience and snowball. Likewise, the sample size is limited and has had a greater representation of women than men. In addition, the sample used in the study had mostly university studies or higher education, came from two-parent families and probably a medium-high socioeconomic status, which makes it a fairly homogeneous sample and makes it difficult to extrapolate results. Finally, the online application of the questionnaires has also prevented access to the study to those who do not have Internet or have difficulties in handling it. Secondly, it must be taken into account the transversal and correlational nature of the data, which do not allow us to indicate cause-effect relationships between the different variables studied, so longitudinal studies are needed to evaluate the direction of the effect between family functioning, differentiation of self and trait anxiety. Finally, another limitation of this study is the use of self-report measures. On the one hand, this way of measuring the variables does not control the possible bias derived from social desirability. On the other hand, the information received about the family functioning of an entire system is provided by the perspective of a single member, which could imply that it does not adjust to reality. Therefore, for future research it would be appropriate to have the perspective of other family members, as well as external observers to obtain more accurate information about the family functioning.

Despite the limitations of this study, the results are consistent with some of the theoretical notions posited by Bowen as the relationship between family dynamics and differentiation of self and between these and anxiety. It also provides new data to the growing literature that confirms the relevance that the construct differentiation of self has transculturally [31], when

observing that these results are similar to those found in other cultures that differ in some characteristics with the Spanish culture. Furthermore, not only at the theoretical level, but also at the practical level, this study points out the importance of the Systemic Theory, since it provides data about the influence of the family of origin and the level of differentiation of self on the adequate psychological and emotional development of the individuals and therefore how it is important to work not only with the individual but also with the family when faced with people who suffer from psychological issues such as anxiety. Finally, this work also provides new information on the possible mediating role played by differentiation of self in the relationship between family functioning and anxiety and the importance of being able to work in the clinic to increase the levels of differentiation and thus improve the psychological well-being of individuals.

For future research, it would be interesting to check if the results obtained in this research are also confirmed in clinical practice. That is, if a therapy is carried out focused on increasing the level of self-differentiation of patients, there is a decrease in aspects related to psychological well-being such as anxiety.

## Supporting information

**S1 Data.**
(SAV)

## Author Contributions

**Conceptualization:** Blanca Dolz-del-Castellar, Jesús Oliver.

**Data curation:** Blanca Dolz-del-Castellar.

**Formal analysis:** Blanca Dolz-del-Castellar.

**Investigation:** Blanca Dolz-del-Castellar.

**Methodology:** Blanca Dolz-del-Castellar, Jesús Oliver.

**Software:** Blanca Dolz-del-Castellar.

**Supervision:** Jesús Oliver.

**Validation:** Jesús Oliver.

**Writing – original draft:** Blanca Dolz-del-Castellar.

**Writing – review & editing:** Jesús Oliver.

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
