## [Decision Letter · Decision Letter 0]

8 Nov 2020

PONE-D-20-20447

Relationship between Family Functioning, Differentiation of Self and Anxiety in Young Adults

PLOS ONE

Dear Dr. Pece,

Thank you for submitting your manuscript to PLOS ONE. After careful consideration, we feel that it has merit but does not fully meet PLOS ONE’s publication criteria as it currently stands. Therefore, we invite you to submit a revised version of the manuscript that addresses the points raised during the review process.

Please address all comments made by the two reviewers and in particular about the statistical issues raised in these two referee reports. Please pay particular attention to issues related to sample size. 

We look forward to receiving your revised manuscript.

Sincerely,

Yann Benetreau, PhD

Senior Editor, *PLOS ONE*

Journal Requirements:

Additional Editor Comments (if provided):

Reviewers' comments:

Reviewer's Responses to Questions

**Comments to the Author**

1. Is the manuscript technically sound, and do the data support the conclusions?

Reviewer #1: Partly

Reviewer #2: Partly

2. Has the statistical analysis been performed appropriately and rigorously? 

Reviewer #1: N/A

Reviewer #2: No

3. Have the authors made all data underlying the findings in their manuscript fully available?

Reviewer #1: Yes

Reviewer #2: Yes

4. Is the manuscript presented in an intelligible fashion and written in standard English?

Reviewer #1: Yes

Reviewer #2: Yes

5. Review Comments to the Author

Reviewer #1: In the manuscript entitled "Relationship between Family Functioning, Differentiation of Self and Anxiety in Young

Adults" the authors examined relations between study variables along with diference tests and a mediation test. Although the manuscript seems interesting and has the potential for the family therapy literature, I have some major concerns about the manuscript:

1. My biggest concern is about the small sample size for these multiple analyses. The authors used scales with 74 items, 20 items, and 40 items in addition to a demographic questionnaire. Along with a lot of items to be filled, they used multiple analyses, which makes the generability of the results very very limited. I recommend authors to enlarge their sample.

2. My second biggest concern is that the authors do not explain their analyses procedures within the manuscript. For instance, how did they perform mediation analysis? Which procedures did they follow? Yes, they say that thay used macro PROCESS version 3.3for SPSS, but what about the procedures? They should explicitly tell their analytic procedures step by step with the relevant citations for each methodology they used.

3. The authors should give more background information for the rationale of their study. In this form, they explain each variable separately, but the readers should know how these constructs are related. What are the potential mechanisms bringing these constructs together in a research? Thus, the authors should give these information instead of giving seperate construct explanations.

Reviewer #2: Important note: This review pertains only to ‘statistical aspects’ of the study and so ‘clinical aspects’ [like medical importance, relevance of the study, ‘clinical significance and implication(s)’ of the whole study, etc.] are to be evaluated [should be assessed] separately/independently. Further please note that any ‘statistical review’ is generally done under the assumption that (such) study specific methodological [as well as execution] issues are perfectly taken care of by the investigator(s). This review is not an exception to that and so does not cover clinical aspects {however, seldom comments are made only if those issues are intimately / scientifically related & intermingle with ‘statistical aspects’ of the study}. Agreed that ‘statistical methods’ are used as just tools here, however, they are vital part of methodology [and so should be given due importance].

COMMENTS: Your ABSTRACT is well drafted but assay type. Please note that it is preferable [though this article type is ‘Research Article’] to divide the ABSTRACT with small sections like ‘Objective(s)’, ‘Methods’, ‘Results’, ‘Conclusions’, etc. [refer to item 1b of CONSORT checklist 2010: Structured summary of trial design, methods, results, and conclusions] which is an accepted practice of most good/standard journals [including PLOS-ONE]. It will definitely be more informative then, I guess, whatever the article type may be.

Though measures/tools used are appropriate, most of them yield data that are in [at the most] ‘ordinal’ level of measurement [and not in ratio level of measurement for sure {as the score two times higher does not indicate presence of that parameter/phenomenon as double (for example, a Visual Analogue Scales VAS score or say ‘depression’ score)}]. The Family Adaptability and Cohesion Evaluation Scale-20Esp (FACES - 20Esp), the Differentiation of Self Scale (EDS) and the State-Trait Anxiety Inventory (STAI) are no exception. Application/use of suitable non-parametric test(s) is/are indicated/advisable [even if distribution may be ‘Gaussian’ (i.e. normal)] in such cases.

Remember that “Inferential statistics (i.e. hypothesis testing + estimation of CI) is built on the population model (i.e. the underlying assumption is that there is a population and we are dealing with random sample(s) drawn from that population). Although in clinical trial (involving at least two groups) we do not really deal with random samples (generally a non-probabilistic convenience sampling), ‘allocation’ to treatment groups is ‘randomly’ done which enable us to evoke the population model and we can use inferential statistics safely. But when there is only one group (so that there is no question of random allocation), with ‘non-random’ selection, it may be questionable to use inferential statistics. For a pilot study it is alright to have ‘single-arm design’, or when that is the only possibility’. By quoting this I am not suggesting ‘Do not use any inferential statistics here’, however, it is very essential to keep the limitations in mind while interpreting results.

Further note that highly significant (large) value of ‘Pearson’s correlation coefficient’ alone does not imply cause-effect relationship. There are other certain criteria which are to be considered before making any causal inference(s). Statistical test usually used to assess significance of Pearson’s ‘Correlation coefficient (r)’ is ‘t’ [where t = { r √ [(n-2) / (1-r2)] }for df=n-2, n is sample size] and here Ho is that the population/standard value of ‘r’ is zero. You need r=0.878 to be significant at 5% when n=5 but you need r=0.273 if n=50 & you need only r=0.088 if n=500. ‘P-value’ heavily depends on sample size. Therefore, it is customary to use the (available in most text books on ‘Biostatistics’ or on ‘www/net’) guidelines for interpreting positive or negative correlations (and do not rely only on corresponding ‘P’-value but also consider an absolute value of ‘Correlation coefficient’).

In addition, also note that ‘Logistic regression’ is (though semi-parametric) considered as non-parametric [for all practical purposes]. Therefore, many times it is preferred oven ‘Simple/Multiple Linear Regression’ {because often few ‘independent’ variables are not continuous as required (strictly assumed as continuous) by ‘Multiple Linear Regression’}. Completely (pure/exclusive) non-parametric regression though available, is not very popular [and so software is not available for estimation (in my knowledge)]. Therefore, not necessary to go for ‘non-parametric regression’ and ‘Logistic regression’ is enough to apply here. However, note that [as pointed out in ‘limitations’ (first limitation) on page 23], this design (cross-sectional survey) do not allow to indicate cause-effect relationships between the different variables studied.

In context of ‘The Differentiation of Self Scale (DSS) given on page 12, please note the following:

Whenever response options ranged from 1=strongly disagree to 4=strongly agree (or ranging from 1 (strongly disagree) to 6 (strongly agree)), while using a ‘Likert’ scale responses, recoding [like strongly disagree=-2, disagree=-1, neutral=0, agree=1, strongly agree=2] may yield correct and meaningful ‘arithmetic mean’ which is useful not only for comparison but has absolute meaning, in my opinion. Application of any statistical test(s) assume that meaning of entity used (mean, SD, etc) has a particular meaning. However, ‘α’ [alpha] or most other measures of reliability/correlation will remain same.

I believe that the ‘F’ value given in table-2 [Table 2: Difference of Means between Women and Men in the Differentiation of Self subscales] is from multivariate analysis of variance (MANOVA) as said in text. When only two groups are to be compared {ex. Women and Men}, we use ‘t’ test for two independent groups [non-parametric equivalent to unpaired ‘t’ test is Mann-Whitney ‘U’ test] and not ANOVA ‘F’ [definitely/certainly not ‘multivariate’ ANOVA called ‘MANOVA’]. Although ‘F’ and ‘t’ are mathematically related/equivalent [square of ‘t’ is exactly equal to ‘F’ if (mistakenly) calculated for two groups], logic/philosophy (and so underlying assumptions) behind their development and algorithms used for estimation of test statistic are different. They are applicable in different situations.

[in table-2, for ‘total’ mean=4.17, SD=0.49, n=123 for ‘Women’ group and mean=4.33, SD=0.41, n=62 for ‘Men’ group then ‘t’=2.21. Repored is F=7.190 (which is unfortunately not just square of 2.21, may be because this ‘F’ is yielded by ‘multivariate MANOVA’ and not ‘univariate ANOVA’). Mind you that this is a scientific/academic document and so should be reported only scientifically correct, appropriate things.]

Moreover, what is ‘Lambda’ in table-2. Interpretation of ‘Lambda=0.833’ is not given anywhere. You may know that Wilks’ lambda is ‘pooled ratio of error variances to effect variance plus error variance’. My question is “what is the role of ‘Lambda’ value here”? Simillarly, I could not understand [refer to second ‘limitation’ on page 23] “When you said ‘the sample has had a greater representation of women than men, which has made it difficult to perform analysis of comparison of means between groups” How was that? Please explain.

Overall the study has potential but needs ‘major revision’, in my opinion. Introduction and description/explanation of other concepts [like Differentiation of Self] are nice.

6. PLOS authors have the option to publish the peer review history of their article (what does this mean?). If published, this will include your full peer review and any attached files.

Reviewer #1: No

Reviewer #2: No

---

## [Author Response · Author response to Decision Letter 0]

19 Nov 2020

Comments to the Author

1. Is the manuscript technically sound, and do the data support the conclusions?

Reviewer #1: Partly

Reviewer #2: Partly

The article insists that the results are limited to our Spanish sample and it indicates as a limitation the small sample size. Nevertheless, after the new revision, we think that the study has been conducted rigorously, contains interesting results and the discussion points out that the results coincide with Bowen´s postulates and with those obtained in other studies carried out in other countries as well as in Spain. 

2. Has the statistical analysis been performed appropriately and rigorously? 

Reviewer #1: N/A

Reviewer #2: No

Taking into account your suggestion, we substituted the MANOVA test for several Student´s t test. After the new revision, we think that all the statistical analysis have been performed appropriately and rigorously.

3. Have the authors made all data underlying the findings in their manuscript fully available?

Reviewer #1: Yes

Reviewer #2: Yes

4. Is the manuscript presented in an intelligible fashion and written in standard English?

Reviewer #1: Yes

Reviewer #2: Yes

5. Review Comments to the Author

Reviewer #1: 

In the manuscript entitled "Relationship between Family Functioning, Differentiation of Self and Anxiety in Young Adults" the authors examined relations between study variables along with diference tests and a mediation test. Although the manuscript seems interesting and has the potential for the family therapy literature, I have some major concerns about the manuscript:

1. My biggest concern is about the small sample size for these multiple analyses. The authors used scales with 74 items, 20 items, and 40 items in addition to a demographic questionnaire. Along with a lot of items to be filled, they used multiple analyses, which makes the generability of the results very very limited. I recommend authors to enlarge their sample.

Thank you very much for your recommendation. We agree with you but, unfortunately, we do not have the possibility to enlarge the sample and do all the analyzes again, because we have to submit the article by December 23th and we are really busy with other research projects underway and with the bimodal teaching at the university due to the pandemic in Spain. Due to the small sample size, we have insisted throughout the article that the results are limited to our Spanish sample and we have not tried to generalize them to the Spanish population. Nevertheless, we have indicated in the discussion that our results coincide with Bowen´s postulates and with those obtained in other studies carried out in other countries and in Spain. 

2. My second biggest concern is that the authors do not explain their analyses procedures within the manuscript. For instance, how did they perform mediation analysis? Which procedures did they follow? Yes, they say that thay used macro PROCESS version 3.3for SPSS, but what about the procedures? They should explicitly tell their analytic procedures step by step with the relevant citations for each methodology they used.

Thank you very much for your recommendation. We have added a new section about Data analysis, where we explain that the assumptions of each test were checked, and we describe all the statistical analyses carried out in the study, and specifically, the mediation analysis and some citations that explain our procedure.

3. The authors should give more background information for the rationale of their study. In this form, they explain each variable separately, but the readers should know how these constructs are related. What are the potential mechanisms bringing these constructs together in a research? Thus, the authors should give these information instead of giving seperate construct explanations.

Thank you very much for your recommendation. We have included in the section Differentiation of self three paragraphs about Bowen´s postulates, which posit that family functioning, differentiation of self and anxiety are related. The studies about these relationships are not abundant, so we have been able to include just a few references (e.g., Chung & Gale, 2009; Duch-Ceballos et al., 2020; Isik & Bulduk, 2015; Johnson et al., 2001; Kim et al., 2014, 2015; Peleg-Popko, 2002a, 2005; Skowron & Friedlander, 1998). However, this is an important reason to perform this study.

Reviewer #2: 

Important note: This review pertains only to ‘statistical aspects’ of the study and so ‘clinical aspects’ [like medical importance, relevance of the study, ‘clinical significance and implication(s)’ of the whole study, etc.] are to be evaluated [should be assessed] separately/independently. Further please note that any ‘statistical review’ is generally done under the assumption that (such) study specific methodological [as well as execution] issues are perfectly taken care of by the investigator(s). This review is not an exception to that and so does not cover clinical aspects {however, seldom comments are made only if those issues are intimately / scientifically related & intermingle with ‘statistical aspects’ of the study}. Agreed that ‘statistical methods’ are used as just tools here, however, they are vital part of methodology [and so should be given due importance]. 

COMMENTS: Your ABSTRACT is well drafted but assay type. Please note that it is preferable [though this article type is ‘Research Article’] to divide the ABSTRACT with small sections like ‘Objective(s)’, ‘Methods’, ‘Results’, ‘Conclusions’, etc. [refer to item 1b of CONSORT checklist 2010: Structured summary of trial design, methods, results, and conclusions] which is an accepted practice of most good/standard journals [including PLOS-ONE]. It will definitely be more informative then, I guess, whatever the article type may be.

Thank you very much for your recommendation. We have included the abstract with all these sections.

Though measures/tools used are appropriate, most of them yield data that are in [at the most] ‘ordinal’ level of measurement [and not in ratio level of measurement for sure {as the score two times higher does not indicate presence of that parameter/phenomenon as double (for example, a Visual Analogue Scales VAS score or say ‘depression’ score)}]. The Family Adaptability and Cohesion Evaluation Scale-20Esp (FACES - 20Esp), the Differentiation of Self Scale (EDS) and the State-Trait Anxiety Inventory (STAI) are no exception. Application/use of suitable non-parametric test(s) is/are indicated/advisable [even if distribution may be ‘Gaussian’ (i.e. normal)] in such cases.

Thank you very much for your comment. We have followed the instructions of the instruments and they do not use standardized scores or percentiles. Also, we have followed the method used in most of the studies about family functioning and differentiation of self.

Remember that “Inferential statistics (i.e. hypothesis testing + estimation of CI) is built on the population model (i.e. the underlying assumption is that there is a population and we are dealing with random sample(s) drawn from that population). Although in clinical trial (involving at least two groups) we do not really deal with random samples (generally a non-probabilistic convenience sampling), ‘allocation’ to treatment groups is ‘randomly’ done which enable us to evoke the population model and we can use inferential statistics safely. But when there is only one group (so that there is no question of random allocation), with ‘non-random’ selection, it may be questionable to use inferential statistics. For a pilot study it is alright to have ‘single-arm design’, or when that is the only possibility’. By quoting this I am not suggesting ‘Do not use any inferential statistics here’, however, it is very essential to keep the limitations in mind while interpreting results. 

Thank you very much for your comment. We have tried to be very cautious with the interpretations. So, we have pointed out in the article that the results are limited to our Spanish sample and we highlight as a limitation that the sample size of our study is limited.

Further note that highly significant (large) value of ‘Pearson’s correlation coefficient’ alone does not imply cause-effect relationship. There are other certain criteria which are to be considered before making any causal inference(s). Statistical test usually used to assess significance of Pearson’s ‘Correlation coefficient (r)’ is ‘t’ [where t = { r √ [(n-2) / (1-r2)] }for df=n-2, n is sample size] and here Ho is that the population/standard value of ‘r’ is zero. You need r=0.878 to be significant at 5% when n=5 but you need r=0.273 if n=50 & you need only r=0.088 if n=500. ‘P-value’ heavily depends on sample size. Therefore, it is customary to use the (available in most text books on ‘Biostatistics’ or on ‘www/net’) guidelines for interpreting positive or negative correlations (and do not rely only on corresponding ‘P’-value but also consider an absolute value of ‘Correlation coefficient’).

Thank you very much for your comment. As we have pointed out in the limitations of the study (on page 26), “It must be taken into account the transversal and correlational nature of the data, which do not allow us to indicate cause-effect relationships between the different variables studied, so longitudinal studies are needed to evaluate the direction of the effect between family functioning, differentiation of self and trait anxiety”. On the other hand, we have interpreted the correlation coefficients, and not the p values, using text books.

In addition, also note that ‘Logistic regression’ is (though semi-parametric) considered as non-parametric [for all practical purposes]. Therefore, many times it is preferred oven ‘Simple/Multiple Linear Regression’ {because often few ‘independent’ variables are not continuous as required (strictly assumed as continuous) by ‘Multiple Linear Regression’}. Completely (pure/exclusive) non-parametric regression though available, is not very popular [and so software is not available for estimation (in my knowledge)]. Therefore, not necessary to go for ‘non-parametric regression’ and ‘Logistic regression’ is enough to apply here. However, note that [as pointed out in ‘limitations’ (first limitation) on page 23], this design (cross-sectional survey) do not allow to indicate cause-effect relationships between the different variables studied.

Thank you very much for your comment. We absolutely agree with this comment. As you have stated, we have pointed out in the limitations of the study, “It must be taken into account the transversal and correlational nature of the data, which do not allow us to indicate cause-effect relationships between the different variables studied, so longitudinal studies are needed to evaluate the direction of the effect between family functioning, differentiation of self and trait anxiety”.

In context of ‘The Differentiation of Self Scale (DSS) given on page 12, please note the following:

Whenever response options ranged from 1=strongly disagree to 4=strongly agree (or ranging from 1 (strongly disagree) to 6 (strongly agree)), while using a ‘Likert’ scale responses, recoding [like strongly disagree=-2, disagree=-1, neutral=0, agree=1, strongly agree=2] may yield correct and meaningful ‘arithmetic mean’ which is useful not only for comparison but has absolute meaning, in my opinion. Application of any statistical test(s) assume that meaning of entity used (mean, SD, etc) has a particular meaning. However, ‘α’ [alpha] or most other measures of reliability/correlation will remain same.

Thank you very much for your interesting comment. We have tried to follow the instructions of the instruments and the method used in most of the studies about family functioning and differentiation of self., but we will follow your suggestion in future research.

I believe that the ‘F’ value given in table-2 [Table 2: Difference of Means between Women and Men in the Differentiation of Self subscales] is from multivariate analysis of variance (MANOVA) as said in text. When only two groups are to be compared {ex. Women and Men}, we use ‘t’ test for two independent groups [non-parametric equivalent to unpaired ‘t’ test is Mann-Whitney ‘U’ test] and not ANOVA ‘F’ [definitely/certainly not ‘multivariate’ ANOVA called ‘MANOVA’]. Although ‘F’ and ‘t’ are mathematically related/equivalent [square of ‘t’ is exactly equal to ‘F’ if (mistakenly) calculated for two groups], logic/philosophy (and so underlying assumptions) behind their development and algorithms used for estimation of test statistic are different. They are applicable in different situations.

[in table-2, for ‘total’ mean=4.17, SD=0.49, n=123 for ‘Women’ group and mean=4.33, SD=0.41, n=62 for ‘Men’ group then ‘t’=2.21. Reported is F=7.190 (which is unfortunately not just square of 2.21, may be because this ‘F’ is yielded by ‘multivariate MANOVA’ and not ‘univariate ANOVA’). Mind you that this is a scientific/academic document and so should be reported only scientifically correct, appropriate things.]

Moreover, what is ‘Lambda’ in table-2. Interpretation of ‘Lambda=0.833’ is not given anywhere. You may know that Wilks’ lambda is ‘pooled ratio of error variances to effect variance plus error variance’. My question is “what is the role of ‘Lambda’ value here”? 

Thank you very much for your interesting comment. We thought that MANOVA test was the proper test to check differences in some variables that belong to the same construct. Taking into account your suggestion, we have substituted the MANOVA test for several Student´s t test.

Simillarly, I could not understand [refer to second ‘limitation’ on page 23] “When you said ‘the sample has had a greater representation of women than men, which has made it difficult to perform analysis of comparison of means between groups” How was that? Please explain.

Thank you very much for your comment We are sorry, it was a misunderstanding. We have deleted the sentence “, which has made it difficult to perform analysis of comparison of means between groups”.

Overall the study has potential but needs ‘major revision’, in my opinion. Introduction and description/explanation of other concepts [like Differentiation of Self] are nice.

Thank you very much for your recommendation. We have tried to revise all the manuscript taking into account all your comments. We hope you find it suitable.

---

## [Decision Letter · Decision Letter 1]

2 Dec 2020

PONE-D-20-20447R1

Relationship between Family Functioning, Differentiation of Self and Anxiety in Spanish Young Adults

PLOS ONE

Dear Dr. Oliver

Thank you for submitting your manuscript to PLOS ONE. After careful consideration, we feel that it has merit but does not fully meet PLOS ONE’s publication criteria as it currently stands. Therefore, we invite you to submit a revised version of the manuscript that addresses the points raised during the review process.

Please submit your revised manuscript by December 20, 2020. If you will need more time than this to complete your revisions, please reply to this message or contact the journal office at plosone@plos.org. Please include the following items when submitting your revised manuscript:

We look forward to receiving your revised manuscript.

Kind regards,

Filiberto Toledano-Toledano, Ph.D.

Academic Editor

PLOS ONE

Additional Editor Comments:

Considering that for PLOS ONE the type of manuscript that you have sent is a Research Article. I suggest that the authors rewrite the background or introduction section, focusing on reporting only research results and empirical findings that allow a clear answer to the research question, and that in the background section the authors show empirical evidence of the relationship between family functioning, self and anxiety. Considering that your manuscript sent to PLOS ONE is not a narrative review article of the literature, it is not necessary to include in the background section theoretical aspects such as: Systemic Theory, Circumplex Model, and Bowen’s Family Systems Theory. Or these topics could be included, in the context of the article, but not in the text, because it is an empirical article. Therefore, I suggest that the authors consider including in the introductory section only empirical evidence and recent research results on family functioning, self and anxiety. It is suggested to reduce the length of the introductory section of this manuscript. It is suggested to check the following link of Pubmed in which you can find some examples, to read and citation.

https://pubmed.ncbi.nlm.nih.gov/?term=Filiberto%20Toledano%20Toledano&page=2

Reviewers' comments:

Reviewer's Responses to Questions

**Comments to the Author**

1. If the authors have adequately addressed your comments raised in a previous round of review and you feel that this manuscript is now acceptable for publication, you may indicate that here to bypass the “Comments to the Author” section, enter your conflict of interest statement in the “Confidential to Editor” section, and submit your "Accept" recommendation.

Reviewer #1: All comments have been addressed

Reviewer #2: (No Response)

2. Is the manuscript technically sound, and do the data support the conclusions?

Reviewer #1: Yes

Reviewer #2: (No Response)

3. Has the statistical analysis been performed appropriately and rigorously? 

Reviewer #1: Yes

Reviewer #2: (No Response)

4. Have the authors made all data underlying the findings in their manuscript fully available?

Reviewer #1: Yes

Reviewer #2: (No Response)

5. Is the manuscript presented in an intelligible fashion and written in standard English?

Reviewer #1: Yes

Reviewer #2: (No Response)

6. Review Comments to the Author

Reviewer #1: After reviewing the changes made by the author(s), I can see that they have attended to the revisions appropriately and the manuscript seems acceptable to be published within this form.

Reviewer #2: COMMENTS: Since for most of the comments made on earlier draft by me (and by other respected reviewer also) a general response is “We have tried to follow the instructions of the instruments and the method used in most of the studies about family functioning and differentiation of self [although it is said sometimes that “we will follow your suggestion in future research”], I am confused now about any recommendation. However, I see that the manuscript is improved.

Nevertheless, now I see few different issues/queries: in table-2, ‘what is “d” [last column, no.8]?’ is not explained. I believe, ‘d’ is ‘effect size’, an (useful) index given by Cohen. If so, ‘why it is not described properly?’. At few places kindly check for the ‘English’ language {example, 2nd para below table-2 you say ‘linear regression model was performed ‘. Is model performed? Or applied/used?}. Agreed, that English is not our mother tongue however, remember that this is a scientific/academic document and so all details should be clearly/correctly communicated.

In same para [2nd para below table-2] you have given R²=.188 {by-the-way R² is called as ‘coefficient of Determination’}, note that variation explained (18.8%) is not much. As I said earlier [which unfortunately still holds], “Overall the study has potential but needs ‘major revision’, in my opinion”.

7. PLOS authors have the option to publish the peer review history of their article (what does this mean?). If published, this will include your full peer review and any attached files.

Reviewer #1: No

Reviewer #2: No

---

## [Author Response · Author response to Decision Letter 1]

10 Dec 2020

Additional Editor Comments:

Considering that for PLOS ONE the type of manuscript that you have sent is a Research Article. I suggest that the authors rewrite the background or introduction section, focusing on reporting only research results and empirical findings that allow a clear answer to the research question, and that in the background section the authors show empirical evidence of the relationship between family functioning, self and anxiety. Considering that your manuscript sent to PLOS ONE is not a narrative review article of the literature, it is not necessary to include in the background section theoretical aspects such as: Systemic Theory, Circumplex Model, and Bowen’s Family Systems Theory. Or these topics could be included, in the context of the article, but not in the text, because it is an empirical article. Therefore, I suggest that the authors consider including in the introductory section only empirical evidence and recent research results on family functioning, self and anxiety. It is suggested to reduce the length of the introductory section of this manuscript. It is suggested to check the following link of Pubmed in which you can find some examples, to read and citation.

 Thank you very much for your recommendation. We have tried to reduce the theoretical section, keeping the definition of the variables and the empirical review.

Comments to the Author

1. If the authors have adequately addressed your comments raised in a previous round of review and you feel that this manuscript is now acceptable for publication, you may indicate that here to bypass the “Comments to the Author” section, enter your conflict of interest statement in the “Confidential to Editor” section, and submit your "Accept" recommendation.

Reviewer #1: All comments have been addressed

Reviewer #2: (No Response)

2. Is the manuscript technically sound, and do the data support the conclusions?

Reviewer #1: Yes

Reviewer #2: (No Response)

3. Has the statistical analysis been performed appropriately and rigorously? 

Reviewer #1: Yes

Reviewer #2: (No Response)

4. Have the authors made all data underlying the findings in their manuscript fully available?

Reviewer #1: Yes

Reviewer #2: (No Response)

5. Is the manuscript presented in an intelligible fashion and written in standard English?

Reviewer #1: Yes

Reviewer #2: (No Response)

6. Review Comments to the Author

Reviewer #1: After reviewing the changes made by the author(s), I can see that they have attended to the revisions appropriately and the manuscript seems acceptable to be published within this form.

Reviewer #2: COMMENTS: Since for most of the comments made on earlier draft by me (and by other respected reviewer also) a general response is “We have tried to follow the instructions of the instruments and the method used in most of the studies about family functioning and differentiation of self [although it is said sometimes that “we will follow your suggestion in future research”], I am confused now about any recommendation. However, I see that the manuscript is improved.

 Thank you very much for your comment. We have considered carefully all your advice, which we appreciate very much. But in this case we have tried to follow the instructions of the instruments used in this study and the method of most research about family functioning and differentiation of self, which use parametrical tests. It is true that Likert type scales are technically ordinal because they consist of a series of ordered categories. However, several authors have found consistent support for the use of these variables as approximately continuous. Firstly, they have observed that Likert type scales, or ordinal variables with five or more categories can be used as continuous without any harm to the statistical analysis (Johnson & Creech, 1983; Norman, 2010; Sullivan & Artino, 2013; Zumbo & Zimmerman, 1993). Secondly, they argue that the total scores of the Likert type scales consist of the sums or means of severalquestions or items–ordinal variables-. The sums or the means result in a number of categories much higher than the categories of the ordinal Likert scales they were calculated from, which results in an approximately continuous variable (Norman, 2010). This is why we have performed parametric tests in this study. However, we pointed out that “we will follow your suggestion in future research” because we are constructing a new Likert scale and we will use the range of responses that you have suggested (from -3 to +3), and if most studies about a topic use non-parametric tests, we will conduct them.

References

Johnson, D.R., & Creech, J.C. (1983). Ordinal measures in multiple indicator models: A simulation study of categorization error. American Sociological Review, 48, 398-407.https://doi.org/10.2307/2095231

Norman, G. (2010). Likert scales, levels of measurement and the “laws” of statistics. Advances in Health Sciences Education, 15(5), 625-632. https://doi.org/10.1007/s10459-010-9222-y

Sullivan, G. & Artino, A. R. (2013). Analyzing and Interpreting Data From Likert-Type Scales. Journal of Graduate Medical Education, 5(4), 541-542.https://doi.org/10.4300/JGME-5-4-18

Zumbo, B. D., & Zimmerman, D. W. (1993). Is the selection of statistical methods governed by level of measurement? Canadian Psychology, 34, 390-400.https://doi.org/10.1037/h0078865

Nevertheless, now I see few different issues/queries: in table-2, ‘what is “d” [last column, no.8]?’ is not explained. I believe, ‘d’ is ‘effect size’, an (useful) index given by Cohen. If so, ‘why it is not described properly?’. 

Thank you very much for your comment. We have change the title of the last column, and the results in parentheses from “d” to “Cohen´s d”. We have also interpreted the values of all effect sizes.

At few places kindly check for the ‘English’ language {example, 2nd para below table-2 you say ‘linear regression model was performed ‘. Is model performed? Or applied/used?}. Agreed, that English is not our mother tongue however, remember that this is a scientific/academic document and so all details should be clearly/correctly communicated.

 Thank you very much for your correction. We have tried to write the text properly, but it seems that we have made a mistake. We have changed the sentence from “linear regression model was performed” to “linear regression analysis was conducted”.

In same para [2nd para below table-2] you have given R²=.188 {by-the-way R² is called as ‘coefficient of Determination’}, note that variation explained (18.8%) is not much. As I said earlier [which unfortunately still holds], “Overall the study has potential but needs ‘major revision’, in my opinion”.

 Thank you very much for your comments. We did not intend to find the model that best predicts differentiation of self, but to know if family functioning predicted differentiation of self and what was the predictive power, which represents a fifth of differentiation. However, following your advice, we have decided to add another result that we also found: family functioning and differentiation of self predict 50% of anxiety. We believe that this result is also relevant and may be of greater interest than the previous one.

---------

 Thank you very much for your all your recommendations. We have tried to revise all the manuscript taking into account all your comments. We hope you find it suitable.

---

## [Decision Letter · Decision Letter 2]

28 Jan 2021

Relationship between Family Functioning, Differentiation of Self and Anxiety in Spanish Young Adults

PONE-D-20-20447R2

Dear Dr. Oliver,

We’re pleased to inform you that your manuscript has been judged scientifically suitable for publication and will be formally accepted for publication once it meets all outstanding technical requirements.

Kind regards,

Valsamma Eapen, MBBS, PhD, FRCPsych, FRANZCP

Academic Editor

PLOS ONE

Additional Editor Comments (optional):

Reviewers' comments:

Reviewer's Responses to Questions

**Comments to the Author**

1. If the authors have adequately addressed your comments raised in a previous round of review and you feel that this manuscript is now acceptable for publication, you may indicate that here to bypass the “Comments to the Author” section, enter your conflict of interest statement in the “Confidential to Editor” section, and submit your "Accept" recommendation.

Reviewer #1: All comments have been addressed

Reviewer #2: All comments have been addressed

2. Is the manuscript technically sound, and do the data support the conclusions?

Reviewer #1: Yes

Reviewer #2: Yes

3. Has the statistical analysis been performed appropriately and rigorously? 

Reviewer #1: Yes

Reviewer #2: Yes

4. Have the authors made all data underlying the findings in their manuscript fully available?

Reviewer #1: Yes

Reviewer #2: Yes

5. Is the manuscript presented in an intelligible fashion and written in standard English?

Reviewer #1: Yes

Reviewer #2: Yes

6. Review Comments to the Author

Reviewer #1: After reviewing the changes made by the author(s), I can see that they have attended to the revisions appropriately and the manuscript seems acceptable to be published within this form.

Reviewer #2: COMMENTS: Since all the comments made on earlier draft by me (and hopefully by other respected reviewers also) are attended positively/adequately, I am fully satisfied and the manuscript is improved a lot.

Thank for giving/quoting references on 'Likert type scales, or ordinal variables with five or more categories can be used as continuous without any harm to the statistical analysis'. Despite knowing/aware of most {not all} these articles, I passed those comments {note that recoding I suggested of ‘Likert type scales’ is very much valid} I am of the opinion that authors then discuss [in ‘Methods’ section] as to ‘why they have chosen to portray and analyze their data in a particular way. Reviewers, readers, and especially editors will greatly appreciate this additional effort.

I recommend acceptance.

7. PLOS authors have the option to publish the peer review history of their article (what does this mean?). If published, this will include your full peer review and any attached files.

Reviewer #1: **Yes: **Erkan Işık

Reviewer #2: No

---

## [Editor Report · Acceptance letter]

2 Feb 2021

PONE-D-20-20447R2 

Relationship between Family Functioning, Differentiation of Self and Anxietyin Spanish Young Adults 

Dear Dr. Oliver:

I'm pleased to inform you that your manuscript has been deemed suitable for publication in PLOS ONE. Congratulations! Your manuscript is now with our production department. 

Kind regards, 

on behalf of

Prof. Valsamma Eapen 

Academic Editor

PLOS ONE